# Within-host evolution of SARS-CoV-2 in an immunosuppressed COVID-19 patient as a source of immune escape variants

Sebastian Weigang [1,7], Jonas Fuchs [1,7], Gert Zimmer [2], Daniel Schnepf [1], Lisa Kern[1], Julius Beer[1], Hendrik Luxenburger [3], Jakob Ankerhold[1], Valeria Falcone [1], Janine Kemming[3], Maike Hofmann [3], Robert Thimme [3], Christoph Neumann-Haefelin [3], Svenja Ulferts [4], Robert Grosse [4], Daniel Hornuss [5], Yakup Tanriver[6], Siegbert Rieg[5], Dirk Wagner[5], Daniela Huzly[1], Martin Schwemmle [1], Marcus Panning [1,8✉] & Georg Kochs [1,8✉]

The origin of SARS-CoV-2 variants of concern remains unclear. Here, we test whether intra-host virus evolution during persistent infections could be a contributing factor by characterizing the long-term SARS-CoV-2 infection dynamics in an immunosuppressed kidney transplant recipient. Applying RT-qPCR and next-generation sequencing (NGS) of sequential respiratory specimens, we identify several mutations in the viral genome late in infection. We demonstrate that a late viral isolate exhibiting genome mutations similar to those found in variants of concern first identified in UK, South Africa, and Brazil, can escape neutralization by COVID-19 antisera. Moreover, infection of susceptible mice with this patient's escape variant elicits protective immunity against re-infection with either the parental virus and the escape variant, as well as high neutralization titers against the alpha and beta SARS-CoV-2 variants, B.1.1.7 and B.1.351, demonstrating a considerable immune control against such variants of concern. Upon lowering immunosuppressive treatment, the patient generated spike-specific neutralizing antibodies and resolved the infection. Our results suggest that immunocompromised patients could be a source for the emergence of potentially harmful SARS-CoV-2 variants.

[1] Institute of Virology, Freiburg University Medical Center, Faculty of Medicine, University of Freiburg, Freiburg, Germany. [2] Institute of Virology and Immunology, Bern & Mittelhäusern, Switzerland, and Department of Infectious Diseases and Pathobiology, Vetsuisse Faculty, University of Bern, Bern, Switzerland. [3] Department of Medicine II, Freiburg University Medical Center, Faculty of Medicine, University of Freiburg, Freiburg, Germany. [4] Institute of Experimental and Clinical Pharmacology and Toxicology, Freiburg University Medical Center, Faculty of Medicine, University of Freiburg, Freiburg, Germany. [5] Division of Infectious Diseases, Dept. Med. II, Freiburg University Medical Center, Faculty of Medicine, University of Freiburg, Freiburg, Germany. [6] Division of Nephrology, Dept. Med. IV, Freiburg University Medical Center, Faculty of Medicine, University of Freiburg, Freiburg, Germany. [7] These authors contributed equally: Sebastian Weigang, Jonas Fuchs [8] These authors jointly supervised this work: Marcus Panning, Georg Kochs. ✉email: marcus.panning@uniklinik-freiburg.de; georg.kochs@uniklinik-freiburg.de

ndividuals infected with SARS-CoV-2 develop neutralizing spike-specific antibodies that persist for months and protect against reinfection[1]. Similarly, neutralizing antibodies generated after vaccination efficiently protect from COVID-19[2]. However, the recent emergence of SARS-CoV-2 alpha- (B.1.1.7), beta- (B.1.351), and gamma- (P.1) variants[3–5] pose a global threat due to their increased transmissibility and resistance to neutralizing antibodies[2]. The origin of these virus variants remains unclear, but long-term-infected immunocompromised individuals are a likely source, allowing prolonged viral replication and unhindered adaption to the host[6,7].

In Germany, the SARS-CoV-2 epidemic started with local outbreaks in February 2020. The city of Freiburg at the border to France and Switzerland was a hotspot due to multiple unrecognized infection events in March 2020. Therefore, immunocompromised patients were closely monitored, as these individuals were expected to have an increased risk of developing severe COVID-19 illness and to suffer from long-term persistent infection with prolonged viral shedding.

Here, we characterize the virus genomic changes in an immunosuppressed kidney transplant recipient who acquired SARS-CoV-2 during the early phase of the COVID-19 pandemic. The patient had mild respiratory symptoms and was tested positive for SARS-CoV-2 for over 145 days. During this long period, viruses with multiple amino acid substitutions and deletions in the spike protein evolved. The mutated spike proteins showed increased resistance to neutralizing antibodies, suggesting a partial immune escape. Interestingly, however, a late virus variant isolated from the patient elicited a considerable protective immune response in experimentally infected mice, suggesting that convalescent individuals might become resistant against reinfection by emerging variants of concern.

## Results

### Clinical presentation of the kidney transplant patient persistently infected with SARS-CoV-2.
A 58-year-old male with a history of autosomal dominant polycystic kidney disease was admitted to our university hospital, for renal transplantation performed on March 2020. The patient was treated with a cocktail of tacrolimus, mycophenolate, and prednisone from March until the end of September (Fig. 1a, b).

On March 2020, the patient tested positive for SARS-CoV-2 by reverse transcription-quantitative polymerase chain reaction (RT-qPCR) (day 0, Fig. 1d). The source of infection remained unknown but strict infection prevention measures were initiated. No respiratory symptoms at the time of diagnosis were observed but a CT scan showed mild ground-glass opacities and discrete bilateral pleural effusions on day 4. The patient remained SARS-CoV-2 positive in the following weeks and was therefore kept in isolation. In May, the patient suffered from an acute kidney injury (stage 1) due to a urinary tract infection with *E. coli*, that required antibiotic treatment. The immunosuppressive medication remained unchanged. Furthermore, he was treated for 5-days (day 56 to 60) with Ivermectin (33 mg/day) (Fig. 1c), a broad-spectrum drug with anti-viral activity in cell culture against several viruses including SARS-CoV-2[8,9]. While the bacterial urinary tract infection was controlled, the infection with SARS-CoV-2 was not.

RT-qPCR positive swab samples were used to successfully isolate the virus on VeroE6 cells confirming shedding of infectious SARS-CoV-2 (Fig. 1e). Due to his critical condition, the patient stayed in the hospital until day 72 when he was discharged for home quarantine. However, he was re-hospitalized at day 106 to 126 due to another kidney transplant failure. Afterwards, the immunosuppressive regimen was modified by

withdrawing mycophenolate mofetil treatment and by increasing the dose of prednisone (day 126) to allow for a better antiviral adaptive immune response. After re-admission at day 140 for control purposes, the patient was still RT-qPCR positive. As an attempt to control the infection, he was treated for 10 days with Remdesivir (200 mg on day 140, then 100 mg/daily, day 141–149), a nucleoside analog with anti-SARS-CoV-2 activity in vitro[10] and in vivo[7,11]. Subsequently, negative RT-qPCR tests until day 189 and failed virus isolation attempts suggested that the infection had resolved (Fig. 1d and e) (Supplementary Table 1).

Nucleoprotein (N)-specific antibodies were detected already 12 days after the first positive qPCR result and afterwards surged rapidly (Fig. 1f). In contrast, IgG antibody levels specific for SARS-CoV-2 spike protein (S1) determined by ELISA constantly oscillated around the cutoff value between days 40 and 123. Only, when the patient was hospitalized in August at day 140, high levels of S1-specific IgG were detected and persisted at least until September (day 175) (Fig. 1f). Concomitantly with the increased spike-specific antibodies and the onset of Remdesivir treatment (day 140), RT-qPCR analysis showed steadily increasing Ct values, indicating diminishing virus replication (Fig. 1d and Supplementary Table 1). In summary, during the 25 weeks of infection with SARS-CoV-2 the patient had no severe respiratory or systemic symptoms and was finally able to clear the virus, likely due to the development of neutralizing antibodies and possibly due to the inclusion of antiviral Remdesivir treatment.

### Genetic relationship of the patient´s SARS-CoV-2 variants with circulating strains.
Full-length SARS-CoV-2 genome sequences were obtained from oropharyngeal swabs collected between day 0 and 140 and phylogenetic trees were constructed including sequences representative for the Freiburg area in March 2020 (Fig. 2a) or a set of randomly selected GISAID sequences from Germany between February and April, 2020 (Fig. 2b). The viral genomic sequences obtained from the patient clearly clustered to strains circulating in March in Germany and in the Freiburg area (Fig. 2a, b). The phylogenetic analysis also demonstrated a close relationship with sequences obtained from two patients of our university hospital (Fig. 2a). However, no clear epidemiologic link was found between the immunosuppressed and the other two patients.

Analysis of the viral RNA that had been extracted from the patient samples revealed several nucleotide substitutions in ORF1ab, the spike gene, ORF3a, M, and N genes in comparison to the Wuhan-Hu-1 reference genome (Fig. 2c). The continuous presence of nine common mutations in all sequences argues against possible reinfection but is compatible with viral persistence. Within the first two weeks, no changes in the viral genome were observed, while from day 42 onward acquisition of several mutations occurred. Apart from low-frequency mutations, some mutations accumulated over time are indicative for the selection of distinct variants. The mutation 23403 G, which results in the D614G substitution, marks the B.1-like genotypes that now dominates worldwide[12].

The most remarkable changes found in the S gene, especially in the d105 specimen, which were confirmed by Sanger sequencing, were in-frame deletions and non-synonymous substitutions in the N-terminal domain (NTD) and the receptor-binding motif (RBM) (Fig. 2d), respectively. Interestingly, the two amino acid deletions in the NTD were associated with specific single amino acid substitutions in the RBM: del141-144 with F490L and del244-247 with E484G, suggesting the emergence of at least two variants. Both deletions, which are located in two adjacent flexible loops of the NTD (Fig. 3), might affect the conformation of this

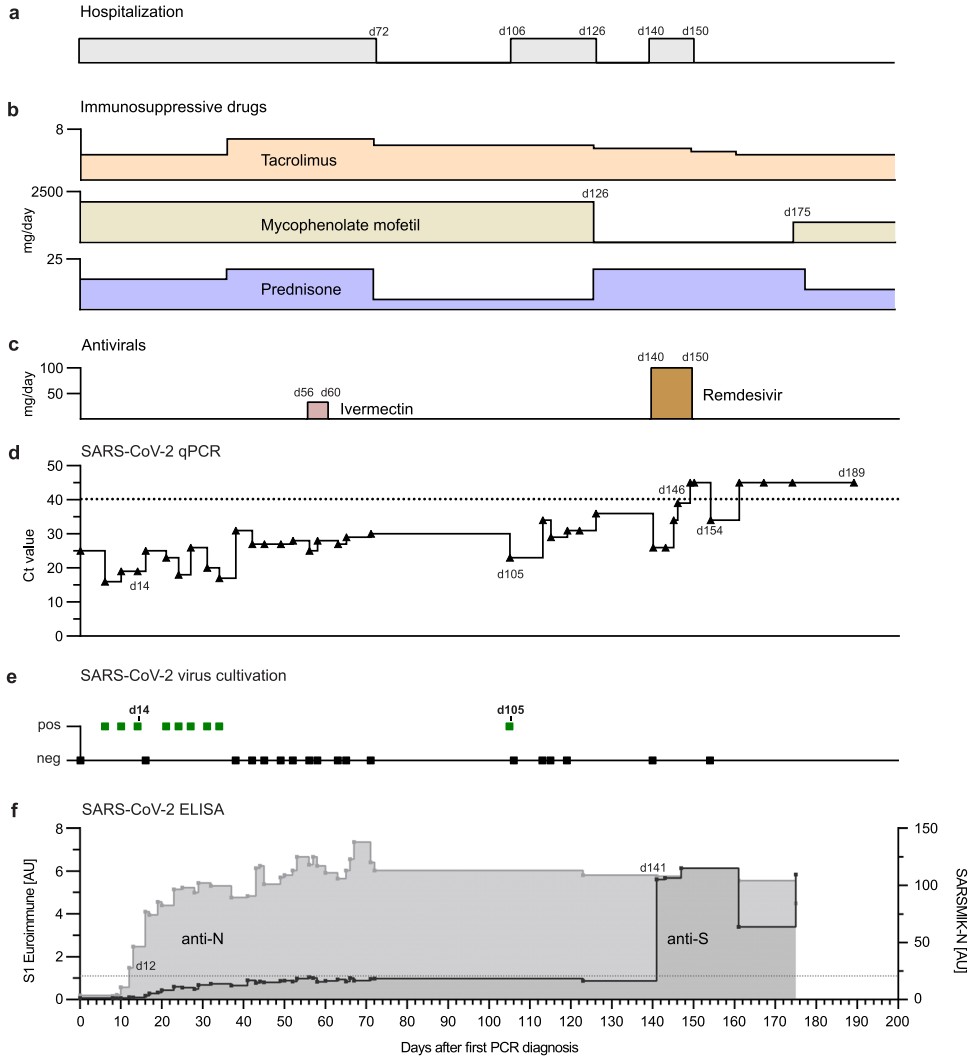

**Fig. 1 Summary of the clinical course of the SARS-CoV-2-positive kidney transplant patient.** Temporal overview of (**a**) hospitalization, (**b**) immunosuppressive treatment (daily dose in mg/day), and (**c**) antiviral therapy (daily dose in mg/day), Remdesivir was given 200 mg on the first day and 100 mg/day 2 to 10. **d** Diagnostic SARS-CoV-2 RT-qPCR cycle threshold (Ct) values of oropharyngeal swabs over time. Day 0 indicates the first positive RT-qPCR result in March 2020, 12 days after kidney transplantation. The dotted line indicates the cutoff value (Ct ≥ 40) between positive and negative results. **e** Attempts of virus isolation from oropharyngeal swabs. **f** Detection of spike S1-subunit- and nucleoprotein (N) specific antibodies by ELISA. The dotted line indicates the anti-S1 ELISA cutoff at 1.1 arbitrary unit (AU).

subdomain and are targets of neutralizing antibodies[13–15]. The two amino acid substitutions (E484G and F490L) may lead to subtle conformational changes in the RBD, which is the main target of neutralizing antibodies and a known hotspot for mutations conferring escape from neutralizing antibodies[16–18]. Thus, the SARS-CoV-2 variants persisting in the immunosuppressed patient share mutations with the escape variants of concern from the UK, South Africa and Brazil (Fig. 2e).

**Prolonged viral persistence did not affect viral fitness.** In the early phase of viral persistence (days 0 to 34) virus isolation was successful, indicating constant virus shedding (Fig. 1e). We repeatedly failed, however, to isolate the virus thereafter when the Ct values increased above 25. On day 105, the Ct value dropped to 23, and virus isolation was again successful (Fig. 1e). The sequences of the two distinct virus isolates obtained at day 14 (d14) and day 105 (d105) were compared to the corresponding sequences obtained from swab samples of the same day. While the sequences of the d14 isolate and the d14 swab were identical, the sequence of the d105 isolate partially differed from the

d105 swab sequence (Fig. 4a). The viral genome sequences of the d105 isolate and the swab samples contained the amino acid deletion 244-247 combined with the E484G mutation, while the amino acid deletion del141-144 and the F490L substitution were only found in the swab samples. Since the majority of the mutations in the d105 isolate was also detected in the swab samples, the d105 virus might represent an abundant genotype that was selected during persistence in the patient. However, we cannot exclude that some alterations were introduced or lost during the process of virus isolation in cell culture (Fig. 4a). For instance, during cultivation in VeroE6 cell culture, we obtained one virus stock of the d105 isolate, designated d105(del), with a deletion of 21 nucleotides (del23601-23621) in about 50% of the spike gene sequences, resulting in an eight amino acids SPRRARSV deletion in the S1/S2 furin cleavage site (Supplementary Fig. 2).

The d14 and d105 isolates both showed accumulation of viral N- and S-proteins in infected cells by indirect immunofluorescence and Western blot analyses comparable to an early SARS-CoV-2 lineage B.1 isolate, Muc-IMB-1[19] (Fig. 4b, c). In VeroE6

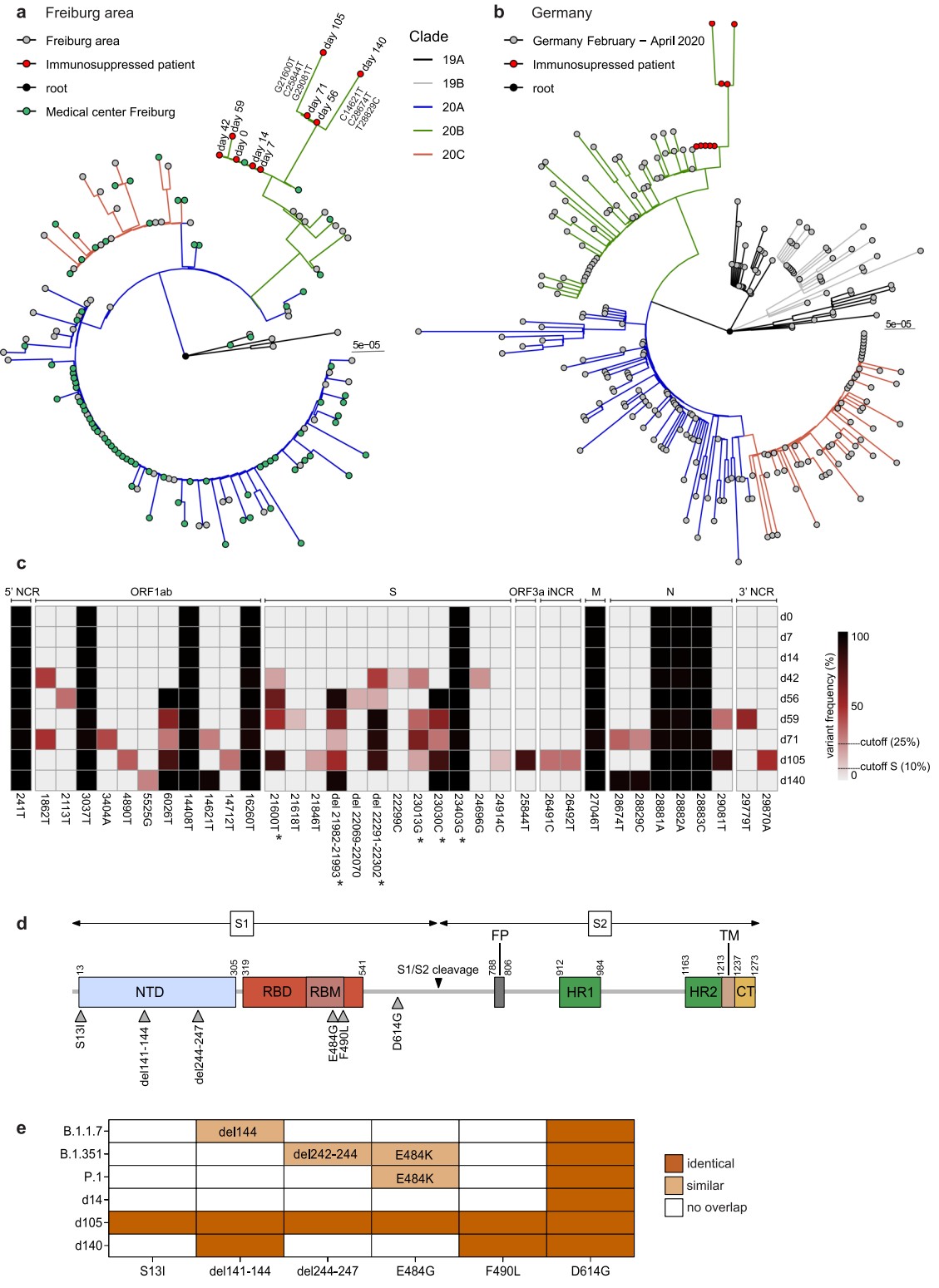

and human lung adenocarcinoma Calu-3 cells, both isolates grew to comparable titers (Fig. 4d, e). To confirm the growth phenotype of d105 in vivo, K18-hACE2 mice encoding the human SARS-CoV-2 receptor, angiotensin-converting enzyme type 2 (ACE2)[20] were intranasally infected with two different doses of the d14 and d105 isolates (200 or 2000 plaque-forming units (pfu)). The infected animals showed a pronounced weight loss within 4 to 8 days and only about 25–30% of the animals infected with 200 pfu survived the infection, whereas all animals

infected with 2000 pfu had to be euthanized due to severe disease symptoms or weight loss (Fig. 4f, g). Together these findings indicate that the mutations in the spike gene and other parts of the d105 genome caused no effect on viral fitness.

However, the VeroE6-derived d105(del) variant showed some attenuation in cell culture and in the K18-hACE2 mice resulting in the survival of most of the infected animals (Supplementary Fig. 2b–e). This attenuation of d105(del) was most likely due to the deletion of the furin cleavage site in the spike as reported before[21].

**Fig. 2 SARS-CoV-2 whole genome sequencing and phylogenetic analysis.** Phylogenetic analysis of the viral sequences obtained from patient swabs between day 0 to day 140, after the first positive RT-qPCR result in March 2020. The sequences were aligned to a set of representative SARS-CoV-2 genome sequences from the Freiburg area (**a**) and from Germany (**b**) between February and April 2020 which have been deposited in the GISAID data bank (Supplementary table 2 and 3). The circularized maximum-likelihood phylogenetic tree was constructed with IQ-Tree (GTR + F + I) and rooted on the Wuhan-Hu-1 reference sequence (NC_045512). The sequences obtained from the immunosuppressed patient are indicated as red dots and lineage-defining mutations are indicated at the respective branches. The scale represents nucleotide substitutions per site. **c** Schematic overview of the viral genome variations from patient swab samples (day 0-140) in comparison to the Wuhan-Hu-1 reference sequence. The heatmap summarizes the positions in the viral genome and the variant frequencies in the different samples (cut off values of 25 and 10% for the S gene, respectively). The days of sampling are indicated at the right and the heatmap color intensity indicates variant frequencies. Stars denote non-synonymous mutations leading to amino acid substitutions in the spike protein (> 50 % of reads). **d** Schematic overview of the SARS-CoV-2 spike protein including the S1 and S2 cleavage products and functional domains such as the N-terminal domain (NTD), receptor-binding domain (RBD), receptor binding motif (RBM), S1/S2 proteolytic furin cleavage site, fusion peptide (FP), heptad repeat regions (HR1/HR2), transmembrane domain (TM) and C-terminal domain (CT). Selected non-synonymous changes in the spike (S) gene from panel **c** are indicated. **e** Summary of mutations found in the spike protein of the patient sequences obtained on d14, d105, and d140 (>50 % of reads) in comparison to circulating new variants of concern: alpha, B.1.1.7[3], beta, B.1.351[4], and gamma, P.1[5].

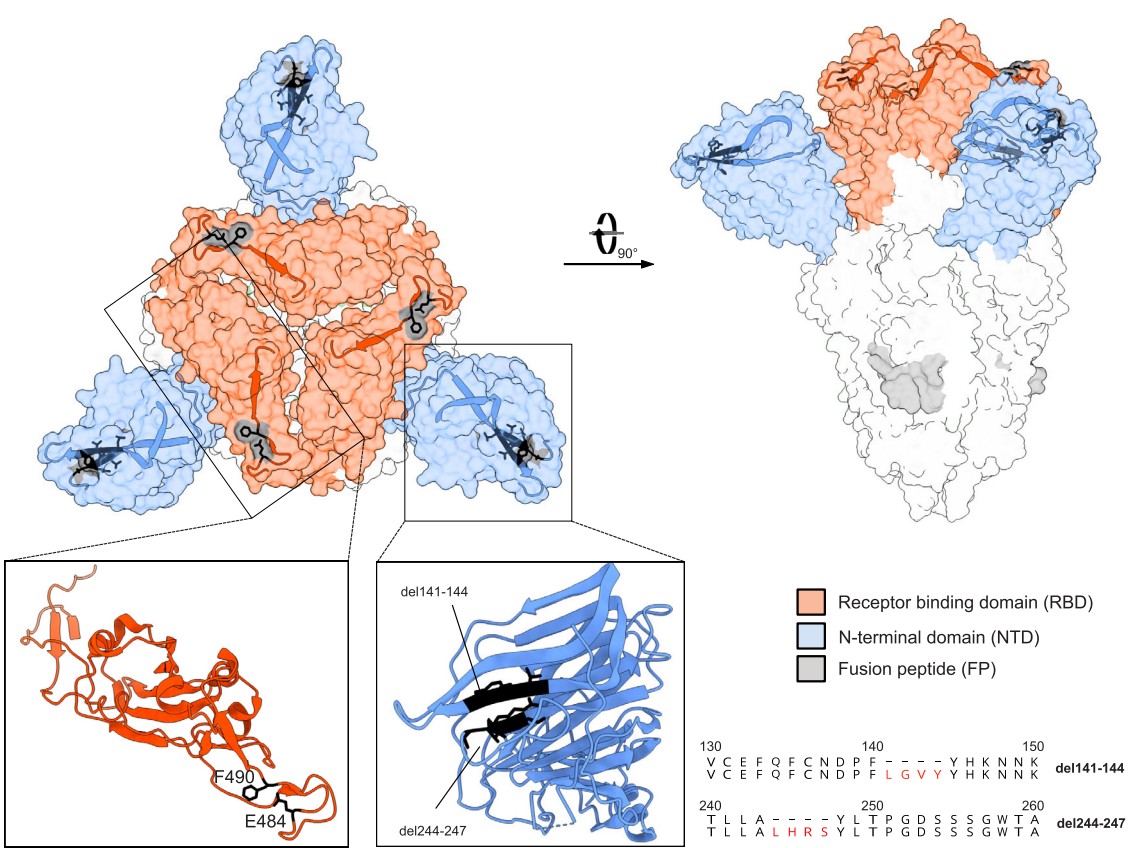

**Fig. 3 Structure of the SARS-CoV-2 spike trimer.** The spike structure (PDB accession number: 7BNM) with the most prominent mutations in the patient viral sequences is shown in the surface presentation. The NTD is colored in blue and the RBD in red. Close-ups of the single NTD and RBD regions defined by boxes are presented as ribbons. The location of the deletions in the NTD and amino acid substitutions in the RBD are indicated by black residues. Furthermore, the deletions in the NTD are displayed as amino acid alignments at the right.

**SARS-CoV-2 escape variant emerged during viral persistence.** The amino acid deletions and substitutions in the spike proteins of the emerging viral variants could have been selected by the antiviral immune response of the host. To address this issue, serum samples of the patient were tested for neutralizing antibodies in a plaque reduction assay performed with either the d14 or the d105(del) virus isolate. Only sera collected from the patient after day 123 showed SARS-CoV-2 neutralizing activity (Fig. 5a), which coincided with the increase of S-specific IgG between day 123 and 140 (Fig. 1f). Intriguingly, while the d14 isolate was efficiently neutralized up to serum dilutions of 1:128, the d105(del) virus was poorly inhibited even at the lowest serum dilutions used (1:32) (Fig. 5a), suggesting that the substitutions in

the d105 spike protein caused an escape from neutralizing antibodies. The neutralization titers detected in sera from the immunosuppressed patient were generally lower than those detected in serum from an immunocompetent convalescent COVID-19 patient (Fig. 5a, positive control). Similarly, antisera of convalescent COVID-19 patients and of individuals previously vaccinated with the BNT162b2 mRNA vaccine (Pfizer/BioNTech) showed higher neutralizing activity against the d14 isolate as compared to the d105(del) isolate (Fig. 5b).

Because of the mutation in the furin cleavage site of the d105(del) spike protein, we re-isolated d105 without changes in the furin cleavage site using Calu-3 cells. This new d105 isolate showed a comparable escape from antibody neutralization

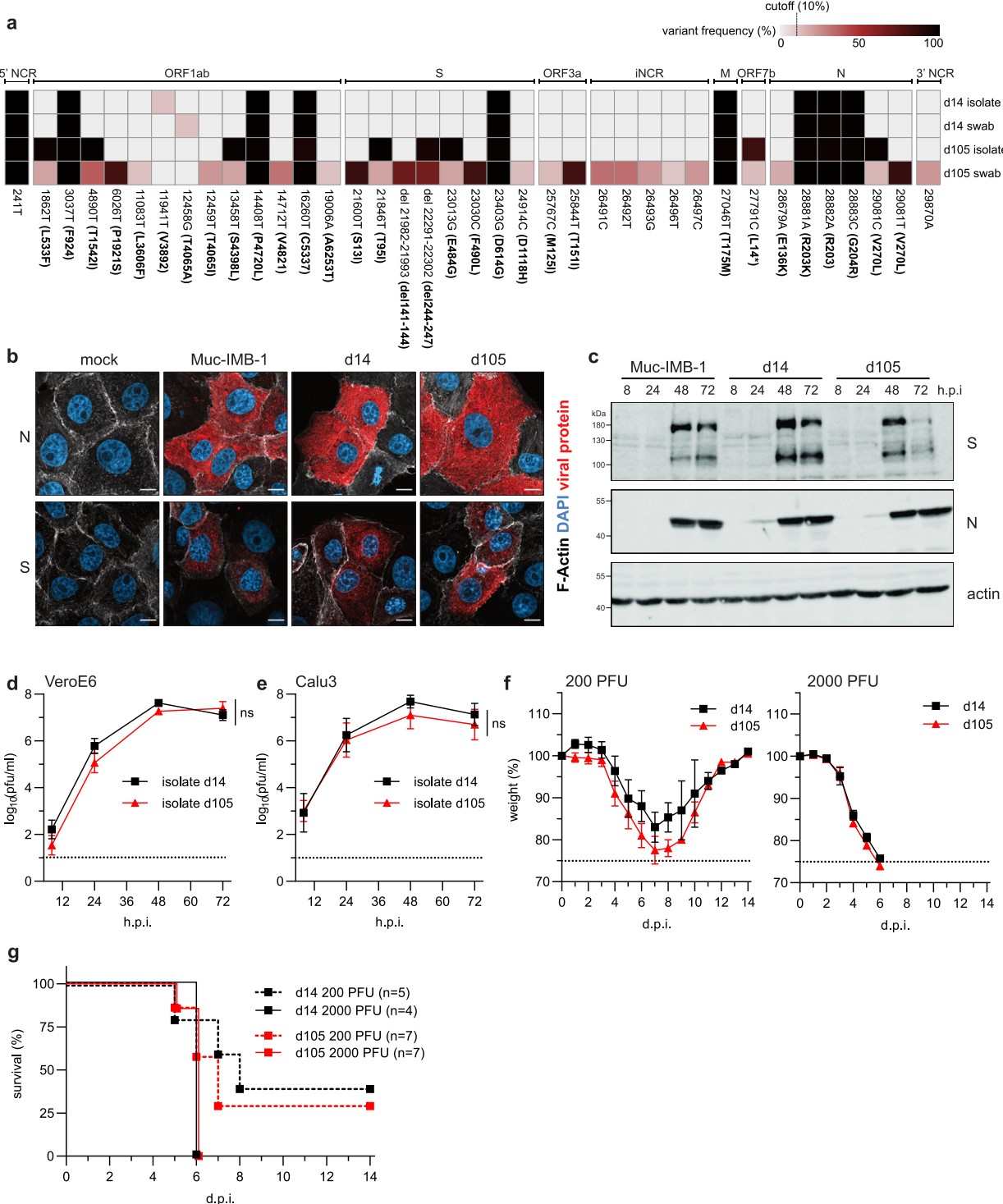

(Supplementary Fig. 3), indicating that the eight amino acids deletion in the spike of d105(del) did not affect the neutralization sensitivity of the d105 spike variant. Of note, not all work could be repeated with this re-isolated d105 virus because samples were expended.

To independently determine the neutralization titer 50 ($NT_{50}$) of convalescent sera against the various spike variants found in the patient, we made use of a virus pseudotype system based on a single-cycle vesicular stomatitis virus (VSV) vector encoding a firefly luciferase reporter, VSV*ΔG(FLuc). High $NT_{50}$ values were observed with the d14 spike protein (Fig. 5c). In contrast,

pseudotype virus bearing the d105 spike protein was neutralized with significantly lower efficacy as indicated by an 8.3-fold reduced $NT_{50}$ value (Fig. 5c).

Next, we analyzed pseudotype virus harboring the d14 spike protein with single or combined mutations including del244-248 and E484G (for d105) and del141-144 and F490L (for d140) (Fig. 2d). Compared to pseudotype virus displaying the parental d14 spike protein, viruses pseudotyped with either the E484G or the F490L mutant spike proteins were equally well neutralized by a COVID-19 convalescent serum, whereas pseudotype virus bearing spike proteins with the amino acid deletions del141-144

**Fig. 4 The late SARS-CoV-2 isolate with mutations in the spike protein does not affect viral fitness. a** Schematic overview of the sequence variations in SARS-CoV-2 genomes detected in early (d14) or late (d105) swab samples and isolated viruses. The heatmap illustrates the positions and the frequency of major variations in the viral genome (cut off 10%). The days of isolation are indicated at the right. The heatmap colors represent the variant frequencies. In ORF7b, L14* indicates a frameshift mutation due to a deletion of two nucleotides. **b** Immunofluorescence analysis of SARS-CoV-2 infected cell cultures. VeroE6 cells were infected with the virus isolates, d14 or d105, and the prototypic lineage B.1 isolate, Muc-IMB-1[19], using 0.1 plaque-forming units (pfu)/cell. At 8 h post-infection, the cells were fixed and stained with SARS-CoV-2 N- and S-specific antibodies (red). In addition, F-actin (white) and nuclear DNA (DAPI, blue) were detected. The scale bar represents 10 µm. **c** Western blot analysis of viral protein expression. Calu-3 cells were infected with 0.001 pfu/cell. Cells were lysed 8 h, 24 h, 48 h, and 72 h post-infection and analyzed using N- and S-specific antibodies. Detection of β-actin was used as a loading control. Panels **b** and **c** show representative data of two independent experiments. **d, e** Growth of the two patient isolates in VeroE6 (**d**) and Calu-3 (**e**) cells infected with the d14 or d105 isolates using 0.001 pfu/cell. At different time points post-infection, cell culture supernatants were collected and viral titers were determined. The log-transformed titers are shown as means ± SD of results from three independent experiments. Significance was determined via two-way ANOVA with a Sidak´s multiple comparison test, **$p < 0.01$, ***$p < 0.001$, ns=non significant. **f** and **g** In vivo infection experiments. Weight loss (**f**) and survival (**g**) of 8 to 12 weeks-old K18-hACE2 mice intranasally infected with 200 pfu of d14 ($n = 5$), 2000 pfu of d14 ($n = 4$), 200 pfu of d105 ($n = 7$) or 2000 pfu of d105 virus ($n = 7$). Signs of disease and body weight loss were monitored daily for 14 days. In panel **f**, data are presented as mean values ± SEM. Source data are provided as a Source Data file.

or del244-247 were significantly less neutralized (Fig. 5d). The combination of the amino acid deletions del141-144 or del244-247 with either E484G or F490L did not further reduce neutralization efficacy. A different pattern was observed when immune serum from a person who had been immunized with an mRNA-based SARS-CoV-2 vaccine was analyzed (Fig. 5e). Using this immune serum, the virus pseudotyped with the E484G mutant spike was less neutralized than the virus bearing the parental spike protein. Furthermore, pseudotype virus displaying spike protein containing both the E484G substitution and the del141-144 deletion showed a reduced $NT_{50}$ value compared to pseudotype viruses containing either E484G or del141-144, suggesting that the two mutations acted in a synergistic manner. In summary, the amino acid changes del141-144 and del244-247, both located in the NTD, and E484G in the RBD all affect crucial antigenic regions[22] which were selected during viral persistence as they allow the escape of SARS-CoV-2 from the humoral immune response.

**SARS-CoV-2 escape variant d105 induces broad protective immunity in vivo**. The reduced neutralization capacity of the patient's sera against the d105 isolate raised the question of whether the changes in the spike protein might have compromised the induction of an efficient antiviral immune response. To address this question, sera from K18-hACE2 mice that survived the infection with the d14 or the d105 isolates (Fig. 4g and Supplementary Fig. 2e) were collected 21 days post-infection or later. In order to increase the number of convalescent sera of animals surviving the infections, we also included sera from K18-hACE2 mice surviving infection with the Muc-IMB-1 isolate, lineage B.1, encoding an identical spike protein sequence like the d14 isolate[19] and sera from mice surviving a d105(del) infection. Due to its attenuated phenotype, the sera of the animals that survived the d105(del) infection had about 2-fold reduced levels of SARS-CoV-2-specific IgG antibodies (assessed by an immunofluorescence-based assay), than the sera of wild type and d105 isolate infected animals (Fig. 6a).

The capacity of sera obtained from the d14- and Muc-IMB-1-infected mice to neutralize the d105 isolate was about 4-fold lower than the neutralizing capacity against the d14 isolate (Fig. 6b), indicating partial immune escape of the d105 isolate. Intriguingly, the d105 mouse sera neutralized the d105 virus more efficiently than the d14 isolate (Fig. 6c). Additionally, we tested the neutralization capacity of the convalescent mouse sera against two recent German isolates of the alpha (B.1.1.7) and the beta (B.1.351) variants of concern. Sera of wild-type-infected mice were more effective in neutralizing the B.1.1.7 virus variant than the B.1.351 variant (Fig. 6d). However, the opposite was observed

using sera from d105-infected animals since the B.1.351 variant demonstrated a higher sensitivity to neutralization by the d105 sera than the B.1.1.7 variant (Fig. 6e). Finally, the convalescent animals, including those infected with Muc-IMB-1, were challenged with a lethal dose (100,000 pfu) of either the d14 or the d105(del) virus isolate one to four months after the first infection. In contrast to the naïve control animals, all convalescent mice survived the challenge infection without any signs of disease or weight loss (Fig. 6f, g), demonstrating that the animals were protected against challenge infection by both virus variants.

**SARS-CoV-2 specific CD8 + T cells are not driving emergence of escape variants**. Finally, we assessed whether the variations in the spike S1 domain also resulted in an escape from the CD8 + T cell response. We performed in silico prediction of CD8 + T cell epitopes within the SARS-CoV-2 S1 domain restricted by the HLA class I alleles of the immunosuppressed COVID-19 patient (HLA-A*02:01, HLA-A*03:01, HLA-B*51:01, HLA-B*56:01). Using ANN4.0 of the Immune Epitope Database website[23] we identified one nonamer peptide within the NTD with good binding properties to HLA-A*02:01 ($S_{133-141}$ FQFCNDPFL) and another nonamer peptide ($S_{142-150}$ GVYYHKNNK) with binding to HLA-A*03:01, both overlapping with del141-144. In addition, we identified a decamer peptide ($S_{240-249}$) with potential binding to HLA-A*02:01 that overlaps with del244-247. Further CD8 + T cell epitope peptides overlapping with the E484G or F490L substitutions could not be predicted. Subsequently, we tested whether the selected peptides represent SARS-CoV-2-specific CD8 + T cell epitopes by incubating PBMCs from the convalescent, immunosuppressed COVID-19 patient on day 233 or from immunocompetent, convalescent, HLA-A*02:01/HLA-A*03:01 positive COVID-19 patients with the peptides for 14 days. However, after stimulation with the selected peptides neither PBMCs from the immunosuppressed nor from the convalescent donors showed any IFNγ-positive CD8 + T cell response whereas a weak IFNγ-positive CD8 + T cell response targeting the non-overlapping epitope HLA-A*03:01/$S_{378-386}$ was detectable in PBMCs isolated from the immunosuppressed patient (Supplementary Fig. 4). This observation indicates that the predicted SARS-CoV-2 peptides that overlap with the mutated S1 regions do not represent CD8 + T cell epitopes. Hence, the mutations within the S1 regions of the d105 and d140 spike proteins were most probably not selected due to CD8 + T cell escape, but due to escape from the humoral response.

**Discussion**

Circulating SARS-CoV-2 variants typically acquire only a few mutations over time which accumulate at a relatively constant

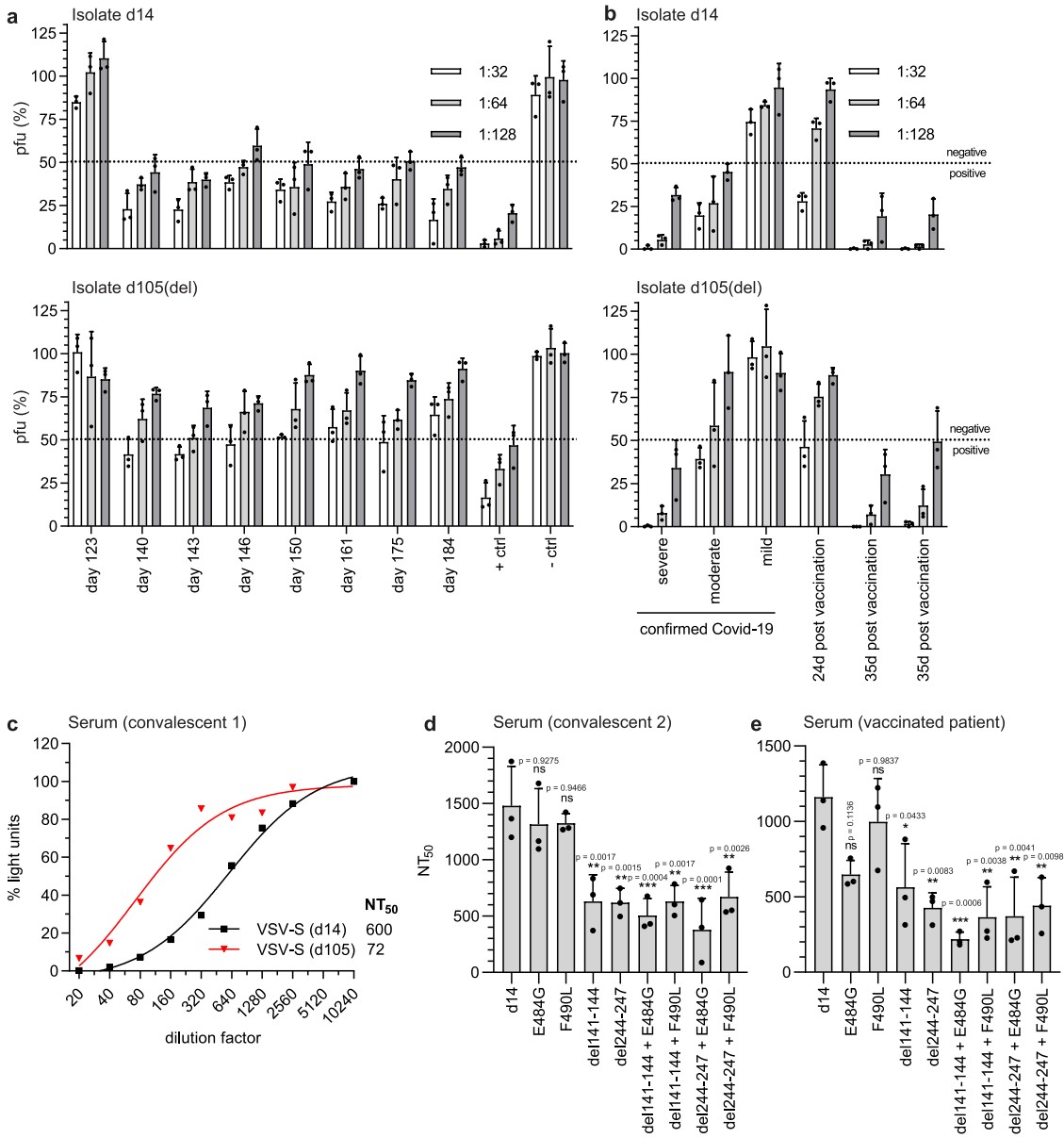

**Fig. 5 Delayed seroconversion and viral escape from the spike protein-specific antibody response. a**, **b** Detection of neutralizing activity of immune sera against SARS-CoV-2 variants. 100 pfu of the d14 and d105(del) isolates were incubated for 60 min at room temperature with serial dilutions of the patient sera. Sera obtained from naïve (– ctrl) or convalescent individuals (+ ctrl) served as negative and positive controls. Virus neutralization was determined by plaque assay on VeroE6 cells. Virus titers are indicated as percentages (mean ± SD) of the titer of the untreated virus inoculum. The dotted lines indicate the cutoff value between positive (<50%) and negative (>50%) neutralization. Shown are the means of three biological replicates. **a** Sera from the immunocompromised patient. The times of blood withdrawal are indicated. **b** Convalescent sera from COVID-19 patients suffering from mild, moderate, or severe disease or human post-vaccination (BNT162b2 mRNA) sera. **c**–**e** Neutralization capacity of SARS-CoV-2 antisera using VSV*ΔG(FLuc) vector pseudotyped with the SARS-CoV-2 spike protein and coding for firefly luciferase. The pseudotyped viruses were incubated with serial dilutions of a COVID-19 convalescent serum prior to inoculation of VeroE6 cells. Pseudotyped virus infection was monitored 16 h post-infection by measuring the firefly luciferase activity in the cell lysates. The control without serum was set to 100%. **c** Neutralization of VSV*ΔG(FLuc) pseudotyped with the early and late SARS-CoV-2 spike variants (d14 and d105). **d**, **e** Neutralization of VSV*ΔG(FLuc) pseudotyped with the d14 spike protein containing the individual or combined mutations found in the late d105 and d140 sequences. Immune sera from two different convalescent COVID-19 patients (**c**, **d**) or a vaccinated person (**e**) were analyzed. The neutralization was determined by calculating the $NT_{50}$ via a non-linear regression (variable slope, four parameters). Shown are means ± SD ($n = 3$). Statistics were calculated with a one-way ANOVA (Tukey's multiple comparison test), ns = non-significant, *$p < 0.05$, **$p < 0.01$, ***$p < 0.001$. The exact $p$-values are given in the figure. Source data are provided as a Source Data file.

rate of about 1-2 mutations per month[24]. Accordingly, the predominant virus genotypes initially found in swab samples during the first weeks of the persistent infection of the immunocompromised patient were relatively stable and grouped into Nextstrain clade 20B, Pangolin lineage B.1.1, together with simultaneously circulating variants (Fig. 2a, b). However, from day 42 onward, synonymous and non-synonymous mutations accumulated in the viral genomes, including two in-frame amino acid deletions in the NTD of the spike protein (del141-144 and del244-247) as well as two single amino acid exchanges (E484G

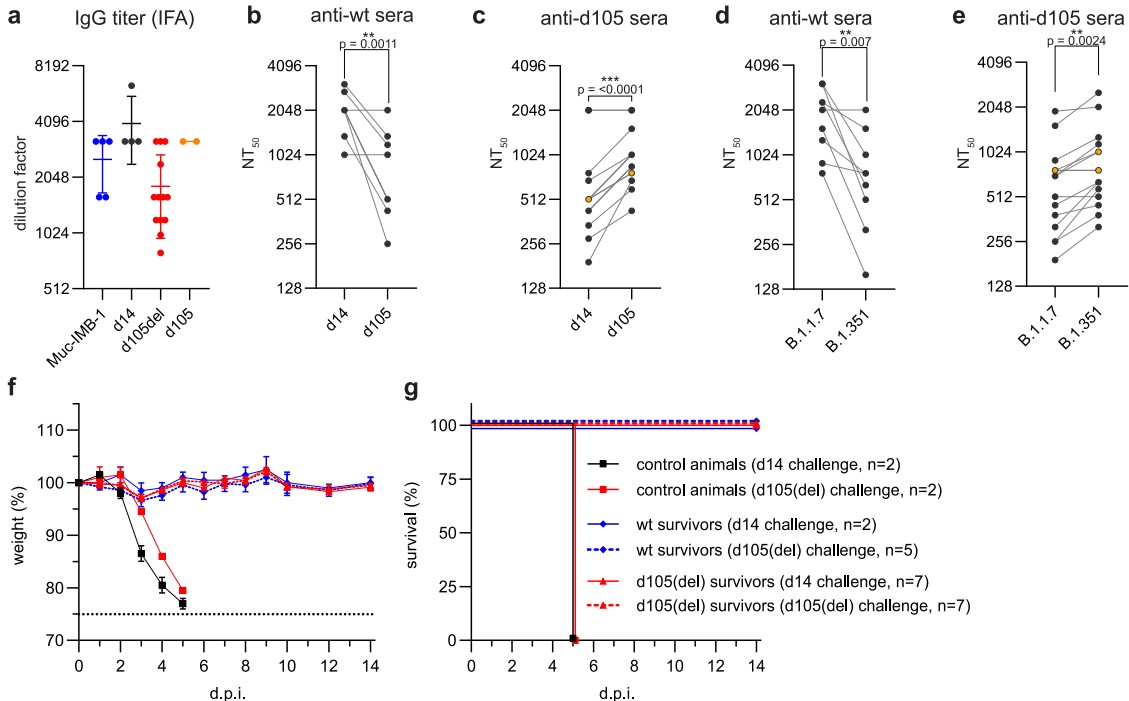

**Fig. 6 Late SARS-CoV-2 d105 isolate elicits cross-reactive protective immunity in mice.** Sera were collected from convalescent K18-hACE2 mice at least 21 days post-infection with Muc-IMB-1 ($n = 5$ blue), d14 ($n = 4$), d105(del) ($n = 13$, red) or d105 ($n = 2$, orange) virus isolates. **a** Anti-SARS-CoV-2 IgG titers of serially diluted sera (mean ± SD) were determined using virus-infected cells and indirect immunofluorescence analysis (IFA). **b, c** Neutralization of d14 and d105 virus isolates by convalescent mouse sera obtained after infection with wild-type SARS-CoV-2, d14 and Muc-IMB-1 (anti-wt sera, $n = 9$) (**b**), or with variant d105 virus isolate (pooled data using anti-d105 ($n = 2$, orange) and anti-d105(del) sera ($n = 13$, black)) (**c**). Neutralization capacity was determined by incubating 400 pfu of either virus isolate with serial dilutions of the mouse sera. The mixture was then applied to VeroE6 cells and infected cells were stained with N-specific antibodies. **d, e** Neutralization of B.1.1.7 and B.1.351 variants of concern by mouse convalescent sera was determined as described in panels **b** and **c**. Neutralization titers, $NT_{50}$, are meant as the highest dilution for each individual serum causing 50% reduction of infectivity. Each serum titer (**b–e**) is shown as the mean out of two independent experiments. Significance was determined via a two-tailed, paired $t$ test with *$p < 0.05$, **$p < 0.01$, ***$p < 0.001$. The exact $p$-values are given in the figure. **f, g** Convalescent animals are protected against re-challenge infection. Weight loss (**f**) and survival (**g**) of convalescent K18-hACE2 mice (mean ± SEM), challenged one to four months after the prime infection. Animals primarily infected with d14 and Muc-IMB-1 viruses (pooled wt survivors, $n = 7$), or with d105(del) virus ($n = 14$) were intranasally challenged with 100,000 pfu of d14 or d105(del) viruses (2–7 mice per group, as indicated). As a control, naïve 8 weeks old K18-hACE2 mice were intranasally infected with 100,000 pfu of d14 or d105(del) isolate viruses ($n = 2$ per group). Source data are provided as a Source Data file.

and F490L) in the RBM. Interestingly, the substitutions in the RBM were exclusively associated with one or the other of the deletions, namely F490L with del141-144 and E484G with del244-247, respectively. This striking interlinked coevolution may have been favored by the necessity to preserve the functionality or stability of the spike protein and to maintain or improve viral fitness. Interestingly, the loops N3 (130-150) and N5 (240-260) that are affected by the two NTD deletions are a preferred target of in-frame deletions of variable length and are therefore referred as "recurrent deletion regions"[14]. To our knowledge, however, further detailed studies upon possible changes in the architecture of the spike proto- or trimer or in the receptor binding affinity by the deletions in the NTP or the exchanges in the RBD found in this study are not available.

Of note, the transient and altering patterns of amino acid changes in the viral spike protein were most likely the result of an ongoing conflict between the persisting virus and the patient's adaptive immune system. The constant but rather weak spike-specific antibody response between days 40 to 123 presumably selected for spike escape variants, as demonstrated by their reduced sensitivity to inhibition by immune sera obtained from the patient at later time points, or from convalescent COVID-19 patients, as well as from COVID-19 vaccinated individuals. It is conceivable that prolonged viral replication in immunosuppressed patients can lead to the emergence of new immune-

escape variants, such as the SARS-CoV-2 variants of concern like the alpha, B.1.1.7, the beta, B.1.351, and the gamma variants, P.1. They all share mutations in the same regions of the spike protein as the escape variants described in this study (Fig. 2e). Accumulation of amino acid substitutions or deletions in similar regions of the spike protein were reported before for isolates of persistently infected, immunosuppressed patients[6,7] and also for isolates from patients treated with antibody cocktails and convalescent plasma[7,16,25].

The reasons for the late but sudden rise of spike-specific antibodies in the patient serum between days 123 and 141 are not clear. We suspect that the pausing of mycophenolate mofetil from day 126 until day 175 favored a broad, spike-specific antibody response that finally terminated the infection. To allow spike-specific antibody production, discontinuation of mycophenolate mofetil treatment of COVID-19-infected transplant recipients is advisable and in line with current clinical guidelines e.g. of the British Transplantation Society (https://bts.org.uk/information-resources/covid-19-information/, updated 22nd January 2021). Adjusting immunosuppressive medications appears to be crucial for the induction of an antiviral immune response and clearance of SARS-CoV-2[26].

Postinfection sera from mice that survived infection with wild-type virus demonstrated reduced neutralizing activity against the d105 virus isolate, highlighting the antibody escape phenotype of

this variant. Conversely, sera from mice previously infected with the d105 isolate more efficiently neutralized the d105 variant than the d14 virus. Hence, broadly neutralizing antibodies that were elicited by new immunogenic epitopes exposed on the mutated d105 spike protein may have controlled the escape variant in the persistently infected patient. To confirm such an extended neutralizing activity of the d105 immune sera, we used recent virus variants of concern and detected enhanced neutralization of the beta variant, B.1.351, by the d105 antisera when compared to the alpha variant, B.1.1.7. These findings match with recent analyses of convalescent plasma samples from patients that recovered from B.1.351 infections. Cele et al. showed efficient neutralization of early 2020 isolates as well as of the late beta variant of concern by these convalescent antisera[27]. Because these globally emerging viruses show a clear escape from vaccine-induced humoral immunity[2,28,29], our findings might be important for the redesign of future vaccines by introducing combinations of mutations into the spike gene that might broaden the specificity of the antiviral immune response.

In summary, we detected SARS-CoV-2 variants in a persistently infected immunocompromised patient, which partially escaped the humoral immune response. Such escape mutants could serve as the initial seed for newly emerging variants with enhanced epidemic potential, especially if they overcome impaired viral fitness by further adaptation. Intriguingly, infections of mice with the late isolate elicited a broadly active neutralizing immune response able to control SARS-CoV-2 variants of concern.

## Methods

**Case history**. A 58-year-old male patient with a history of autosomal dominant polycystic kidney disease (ADPKD) was hospitalized at our university hospital from March until September 2020. He developed the end-stage renal disease in 2014 and required kidney transplantation. Additionally, he suffered from coronary heart disease, arterial hypertension, hyperlipidemia, and obesity. On March 2020, he received a renal transplant from a decreased donor. For induction of immunosuppressive treatment he was given Basiliximab (20 mg, day 0 and day 4 post-transplantation) and Prednisone (250 mg at day 0, 125 mg day 1, 50 mg day 2–5, 20 mg day 6–10, then 15 mg/day). No additional lymphodepleting agent was administered for induction. Moreover and starting on day 0, he received Tacrolimus (10 mg at day 0, 8 mg day 1, 5.5 mg day 2, 5 mg day 3–4, then 4 mg/day) and Mycophenolate mofetil (2000 mg/ day). Maintenance immunosuppression consisted of Tacrolimus (4-6 mg/day), Mycophenolate mofetil (2,000 mg/day), and Prednisone (10–20 mg/day) as indicated in Fig. 1b. The patient received five days of Ivermectin treatment (33 mg/day) according to local guidelines and later the approved maximal dosage of Remdesivir (200 mg on day 1 and 100 mg/day 2–10) according to the guidelines of the European Medicines Agency (EMA).

**Ethics declaration**. The project has been approved by the University Medical Center, Freiburg, ethical committee. Written informed consent was obtained from the patient and the participants providing serum samples. The case study was performed in agreement with principles of the Declaration of Helsinki and CARE guidelines, federal guidelines, and local ethics committee regulations (Albert-Ludwigs-Universität, Freiburg, Germany: No. F-2020-09-03-160428 and no. 322/ 20). All routine virological laboratory testing of patient specimens was performed in the Diagnostic Department of the Institute of Virology, University Medical Center, Freiburg (Local ethics committee no. 1001913).

**Virus detection by qRT-PCR**. SARS-CoV-2 RNA testing of respiratory tract samples was performed using the RealStar SARS-CoV-2 RT-PCR kit (Altona Diagnostics, Hamburg, Germany). RNA samples were extracted using the QIAamp MinElute Virus Spin kit (Qiagen, Hilden, Germany). Tests were performed and interpreted according to the manufacturer's instructions and semi-quantitative results reported in cycle threshold (Ct) values.

**Serological testing**. Convalescent sera of COVID-19 patients and sera from vaccinees after the second dose of the BNT162b2 mRNA vaccine (Pfizer/BioN-Tech) were obtained from the Hepatology-Gastroenterology-Biobank as part of the Freeze-Biobank Consortium at the University Medical Center Freiburg. Written informed consent was obtained from all blood donors prior to inclusion.

SARS-CoV-2 specific anti-spike protein (S1) IgG (Euroimmun, Medizinische Labordiagnostika AG, Lübeck, Germany) and anti-nucleoprotein (N) IgG ELISA (Mikrogen Diagnostik GmbH, Neuried, Germany) were performed according to the

manufacturer's protocol. Results were evaluated semi-quantitatively as arbitrary units (AU) compared to the manufacturer's calibrators.

To determine the SARS-CoV-2-specific antibodies in mouse sera, VeroE6 cells in 96-well plates were infected with the prototypic Muc-IMB-1virus isolate (kindly provided by Roman Woelfel, Bundeswehr Institute of Microbiology[19]). Fixed and permeabilized cells were incubated with dilutions of the post-infectious mouse sera and SARS-CoV-2-specific antibodies were detected by fluorescence-labeled secondary anti-mouse IgG antiserum. The serum dilution giving a clear fluorescence signal in the infected cells was interpreted as positive.

For the SARS-CoV-2 neutralization plaque reduction assay, serial serum dilutions were incubated with 100 plaque-forming units (pfu) for 1 h. The mixture was dispersed on VeroE6 cells in 12-well format and the cells were overlaid with 0.6% Oxoid-agar for 48 h at 37 °C. The fixed cells were stained with Crystal violet. The number of plaques was compared with an untreated control without serum.

For the detection of neutralizing antibodies by indirect immunofluorescence, 400 pfu of SARS-CoV-2 were preincubated with serially diluted serum samples for 1 h and the mixture was used to infect VeroE6 cells in 96-well plates. For each sample, one control without serum was included. Cells were fixed 20 h post-infection and stained with anti-SARS-CoV nucleocapsid (N) rabbit antiserum (#200-401-A50, Rockland Immunochemicals) (dilution 1:1000). The plates were evaluated by fluorescence microscopy. The highest dilution of the serum that showed less than 50% of infected cells compared to a non-reactive control serum was classified as neutralization titer.

**SARS-CoV-2 S1-specific T cell response**. The S1 amino-acid sequence of SARS-CoV-2 (GenBank: MN908947.3) was analyzed for in silico prediction of peptide binding with ANN 4.0 on the Immune Epitope Database website[23] (https://www.iedb.org/). The HLA-A*02:01-restricted 9-mer peptide, $S_{133-141}$ FQFCNDPF$L$, and the HLA-A*03:01-restricted 9-mer peptide, $S_{142-150}$ GVYYHKNNK, both overlapping with del141-144, the HLA-A*02:01-restricted 10-mer peptide, $S_{244-247}$ TLLALHRSYL, overlapping with del244-247, as well as a HLA-A*03:01-restricted 9-mer peptide, $S_{378-386}$ KCYGVSPTK, of the S1 domain were selected and synthesized for further analysis. Subsequently, PBMCs $(1 − 2 \times 10^6)$ of the convalescent immunosuppressed COVID-19 patient and of four HLA-A*02:01/HLA-A*03:01 positive SARS-CoV-2 convalescent immuno-competent donors were stimulated with these peptides (5 µM) and anti-CD28 mAb (0.5 µg ml$^{-1}$, BD Biosciences) and expanded for 14 days in complete RPMI culture medium containing rIL2 (20 IU ml$^{-1}$, Miltenyi Biotec). IFNγ production was assessed after a 5 h re-stimulation with the respective peptide[30]. Flow cytometric analyses were performed on a FACSCanto II (Becton Dickinson) with FACSDiva software version 8.0.1 (Becton Dickinson). Data were analysed with FlowJo 10.6.2 (Treestar).

**Cell culture**. Virus isolation, cell culture, and mouse infection experiments with SARS-CoV-2 were performed under Biosafety Level 3 (BSL3) protocols at the Institute of Virology, Freiburg, approved by the Regierungspraesidium Tuebingen (No. 25-27/8973.10-18 and UNI.FRK.05.16-29). Filtered throat swab samples were inoculated on African green monkey kidney VeroE6 cells (ATCC CRL-1586) $(2 \times 10^6$ cells) in 4 ml DMEM with 2% FCS and incubated at 37 °C and 5% $CO_2$ for 4–6 days until the cytopathic effect was visible. A first virus stock of the d105 isolate passaged on VeroE6 cells, d105(del), showed multiple differences to the genomic sequence in the corresponding patient's swab, including an eight amino acids deletion in the spike protein. Because of this genetic heterogeneity of the d105 isolate, virus stocks of the d14 and d105 variants were generated in Calu-3 cells from plaque purified viruses. The viral genome sequences of the culture supernatants were determined and the viruses stored at −80 °C. Virus titers were determined by plaque assay on VeroE6 cells.

For neutralization assays we used established prototypic isolates: Muc-IMB-1, lineage B.1, kindly provided by Roman Woelfel, Bundeswehr Institute of Microbiology[19]; alpha variant B.1.1.7 (hCoV-19/Germany/NW-RKI-I-0026/2020; ID: EPI_ISL_751799) and beta variant B.1.351 (hCoV-19/Germany/NW-RKI-I-0029/2020; ID: EPI_ISL_803957), provided by Donata Hoffmann and Martin Beer, Friedrich-Loeffler-Institute, Riems.

VeroE6 or human bronchial epithelium Calu-3 cells (ATCC-HTB-55, kindly provided by Markus Hoffmann, Göttingen) in 6-well plates, $1 \times 10^6$ cells, were infected with a moi of 0.001 for 1.5 h. Cells were washed three times with PBS and overlaid with 2 mL DMEM with 2% FCS. The supernatants were taken at different time points after infection and titers were determined by plaque assay. For viral protein analysis, the cells were lysed with a tissue protein extraction reagent (T-PER; Thermo Fisher Scientific), separated by 10% SDS-polyacrylamid gel electrophoresis, and transferred to PVDF membranes. The membranes were stained with the SARS-CoV-2 N-specific (dilution 1:1000), SARS-CoV-2 spike protein (RBD)-specific (200-401-A50 and 600-401-MS8, Rockland) (dilution 1:500) or actin-specific (A5060, Sigma) rabbit antisera (dilution 1:1000). Detection of the primary antibodies was performed with fluorescent-labeled (Li-COR) secondary antibodies.

**Immunofluorescence analysis**. VeroE6 cells seeded on glass coverslips were infected with SARS-CoV-2 isolates at a moi of 0.1. At 8 hours post-infection, cells were fixed in 4% paraformaldehyde, permeabilized with 0.3% Triton X-100 and

blocked in 10% fetal calf serum. SARS-CoV-2 N- and spike-specific primary antibodies (dilution 1:1000 and 1:500, respectively) and AF568-labeled goat-anti-rabbit (Invitrogen, #A11011, 1:400) secondary antibody as well as AF488-labeled Phalloidin (Hypermol, #8813-01, 1:250) were used for staining. The coverslips were embedded in Diamond Antifade Mountant with 4′,6-diamidino-2-phenylindole (DAPI) (ThermoFisher, #P36971). Fluorescence images were generated using an LSM800 confocal laser-scanning microscope (Zeiss) equipped with a 63X, 1.4 NA oil objective, and Airyscan detector and processed with Zen blue software (Zeiss) and ImageJ/Fiji.

**Virus pseudotype VSV*ΔG(FLuc) neutralization assay.** cDNAs encoding the S protein were prepared from oropharyngeal swab samples of the COVID-19 patient obtained at days d14 and d105 and were cloned into the eukaryotic expression vector pCAGGS[31]. Single and double spike mutations were introduced into the pCAGGS-S (d14) construct. BHK-21 cells (ATCC CCL-10) were transfected with the pCAGGS-S plasmids and later inoculated with 5 ffu/cell of VSV*ΔG(FLuc), coding for firefly luciferase, as described[32]. Cells were incubated in a medium containing the monoclonal mAb I1 antibody (ATCC) directed against VSV-G. The supernatants containing the pseudotype viruses were harvested and stored at −70 °C.

The pseudotyped virus neutralization test was performed as described recently[33]. Pseudotyped VSV*ΔG(FLuc) (200 ffu) were preincubated with twofold serial dilutions of convalescent sera in DMEM cell culture medium. The virus/serum mixture was transferred to VeroE6 cells in 96-well plates and incubated for 16 h at 37 °C. Thereafter, the cells were lysed and firefly luciferase activity was determined using ONE-GloTM substrate (Promega) and a GloMax® plate reader (Promega). The reciprocal antibody dilution causing a 50% reduction of the luminescence signal was calculated and expressed as neutralization titer 50% ($NT_{50}$).

**Infection of K18-hACE2 transgenic mice.** Transgenic (K18-hACE2)2Prlmn mice[20] were purchased from The Jackson Laboratory and bred locally. Hemizygous 8-12-week-old animals of both sexes were used in accordance with the guidelines of the Federation for Laboratory Animal Science Associations and the National Animal Welfare Body. Mice were housed at 14 h light/10 h dark cycles and temperatures of ~18–23 °C with 40–60% humidity. All experiments were performed in compliance with the German animal protection law and approved by the animal welfare committee of the Regierungspraesidium Freiburg (permit G-20/91).

Mice were anesthetized using isoflurane and infected intranasally (i.n.) with virus dilutions in 40 µl PBS containing 0.1% BSA. Mice were monitored daily and euthanized if severe symptoms were observed or bodyweight loss exceeded 25% of the initial weight. Serum samples were collected from the vena facialis. SARS-CoV-2 specific antibody titers were determined by indirect immunofluorescence as described above.

**Whole genome sequencing.** cDNA was produced from extracted RNA of oro-pharyngeal swab samples using random hexamer primers and Superscript III (ThermoFisher) followed by a PCR tiling the entire SARS-CoV-2 genome (ARTIC V3 primersets; https://github.com/artic-network/artic-ncov2019). This produced 400 bp long, overlapping amplicons that were subsequently used to prepare the sequencing library. Briefly, the amplicons were cleaned with AMPure magnetic beads (Beckman Coulter). Afterwards the QIAseq FX DNA Library Kit (Qiagen) was used to prepare indexed paired-end libraries for Illumina sequencing. Normalized and pooled sequencing libraries were denatured with 0.2 N NaOH. This 9 pM library was sequenced on an Illumina MiSeq instrument using the 300-cycle MiSeq Reagent Kit v2.

For sequencing of virus stocks produced in cell culture, RNA was extracted with the Quick-RNA Viral Kit (Zymo Research), and paired-end libraries without previous PCR amplifications were prepared using the TruSeq Stranded Total RNA Kit (Illumina). A total of 10 pM library was sequenced on the Illumina MiSeq instrument.

The de-multiplexed raw reads were subjected to a custom Galaxy pipeline, which is based on bioinformatics pipelines on usegalaxy.eu (https://doi.org/10.5281/zenodo.3685264)[34]. The raw reads were pre-processed with fastp (https://www.biorxiv.org/content/10.1101/274100v2) and mapped to the SARS-CoV-2 Wuhan-Hu-1 reference genome (Genbank: NC_045512) using BWA-MEM (https://academic.oup.com/bioinformatics/article/25/14/1754/225615). For datasets, which were produced with the ARTIC v3 protocol, primer sequences were trimmed with ivar trim (https://andersen-lab.github.io/ivar/html/manualpage.html). Variants (SNPs and INDELs) were called with the ultrasensitive variant caller LoFreq (https://academic.oup.com/nar/article/40/22/11189/1152727), demanding a minimum base quality of 30 and a coverage of at least 5-fold. Afterwards the called variants were filtered based on a minimum variant frequency of 10 % and on the support of strand bias. The effects of the mutations were automatically annotated in the vcf files with SnpEff (https://www.tandfonline.com/doi/full/10.4161/fly.19695). Finally, consensus sequences were constructed by bcftools (v.1.10) (https://academic.oup.com/bioinformatics/article/25/16/2078/204688). Regions with low coverage or variant frequencies between 30 and 70 % were masked with Ns. Raw sequencing data have been submitted to the European Nucleotide Archive (https://www.ebi.ac.uk/ena/browser) under the study accession

number: ERP132087. The final consensus sequences have been deposited in the GISAID database (www.gisaid.org) (Supplementary table 2).

**Sanger Sequencing.** To confirm mutations in the viral spike gene, RT-PCR from oral swabs was performed using the primers 5′-GCATGGTACCACCATGTT TGTTTTTCTTGT-3′ and 5′-CTAGCTCGAGTTATTTGCAGCAGGATCC-3′. The PCR product (3791 nt) contains the nucleotide sequence of the SARS-CoV-2 spike gene without the last 54 nucleotides, resulting in a C-terminal deletion of 18 amino acids. The cDNA was either cloned into pCAGGS plasmid using KpnI and XhoI restriction enzymes for later pseudotyping of VSV*ΔG(FLuc), or directly send for Sanger sequencing (Eurofins Genomics) using the following primers: 5′-G TGTTTAAGAATATTGATGG-3′; 5′-AATAGGCGTGTGCTTAGAAT-3′, 5′-G AGATATTTCAACTGAAATC-3′, 5′-ATGTACATTTGTGGTGATTC-3′, 5′-A GAGCTGCAGAAATCAGAGC-3′.

**Phylogenetic and variant analysis.** All available sequences from Germany deposited in GISAID (http://gisaid.org/) between February and April 2020 were downloaded (as of 11th of February 2021) and 250 sequences were randomly subsampled excluding sequences already deposited by the Virology in Freiburg (Supplementary table 3). For the phylogenetic analysis, the sequences were first aligned with MAFFT (v7.45)[35] and a tree was constructed with IQ-Tree (v2.1.2)[36]. The best-fitting substitution model was automatically determined (GTR + F + I) and the tree was calculated with 1000 boot-strap replicates. Branch support was approximated using the Shimodaira–Hasegawa [SH]-aLRT method (1000 replicates). The tree was rooted to the reference sequence NC_045512. The lineage defining mutations (excluding INDELs) were calculated by reconstructing the ancestral sequences with IQ-Tree along the already calculated tree (-asr and -te option) and aligning these to the respective consensus sequences using MAFFT. The clades were classified with the webservers of Nextclade (clades.next-strain.org) and Pangolin (pangolin.cog-uk.io). To visualize the phylogenetic tree a custom R script was written utilizing the ggtree (v2.2.4) (https://academic.oup.com/mbe/article-abstract/35/12/3041/5142656), treeio (v1.12.0) (https://academic.oup.com/mbe/article-abstract/37/2/599/5601621) and ggplot2 (v3.3.3) packages (https://link.springer.com/chapter/10.1007/978-3-319-24277-4_12). An in-house R script was also used to plot the variant frequencies that were detected by LoFreq as a heatmap (pheatmap package v1.0.12). Both scripts are publicly available (github.com/jonas-fuchs/SARS-CoV-2-analyses) and the variant frequency plot has been implemented as a galaxy tool (Variant Frequency Plot on usegalaxy.eu).

**Visualization of the spike protein structure.** The EM structure of the closed conformation of D614G SARS-CoV-2 spike protein was loaded from the protein data bank (DOI: 10.2210/pdb7BNM/pdb) and visualized with UCSF ChimeraX version: 1.1 (2020-09-09).

**Reporting summary.** Further information on research design is available in the Nature Research Reporting Summary linked to this article.

## Data availability

All necessary data and informations are given in the manuscript. A Source data file is provided with this paper. The sequence data are submitted to the GISAID database and are publicly available (Supplementary table 2). Raw sequencing data have been submitted to the European Nucleotide Archive (https://www.ebi.ac.uk/ena/browser) under the study accession number: ERP132087. In silico peptide binding was analyzed with ANN 4.0 on the Immune Epitope Database website (https://www.iedb.org/). Further additional information about the patient will not be shared due to the protection of individuals' privacy. Source data are provided with this paper.

## Code availability

The scripts are publicly available (github.com/jonas-fuchs/SARS-CoV-2-analyses) and the variant frequency plot has been implemented as a galaxy tool (Variant Frequency Plot on usegalaxy.eu).

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

## Acknowledgements
We thank Roman Woelfel, Bundeswehr Institute of Microbiology, Muenchen, for providing prototypic Muc-IMB-1 isolate; Donata Hoffmann and Martin Beer, Friedrich-Loeffler-Institut, Insel Riems, for providing the alpha and beta variant virus isolates; Markus Hoffmann, Goettingen, for the Calu-3 cells; Todd Giardiello, Rockland Immunochemicals, PA, for providing anti-N and anti-S specific rabbit antisera; BioCopy GmbH, Emmendingen, for collaboration on serological testing; Otto Haller, Martin Hölzer, Antoni Wrobel, Volker Thiel and Martin Beer for helpful comments on the manuscript. We like to acknowledge the excellent technical assistance by Nathalie Goeppert, Annette Ohnemus, Valentina Wagner, and the whole team of the diagnostic unit. This work was supported by the Bundesministerium fuer Bildung und Forschung (BMBF) through the Deutsches Zentrum fuer Luft- und Raumfahrt, Germany, (DLR, grant number 01KI2077) to MP, RT and MS and by the Federal State of Baden-Wuerttemberg, Germany, MWK-Sonderfoerdermaßnahme COVID-19/AZ.:33-7533.-6-21/7/2 to MS and AZ.:33-7533-6-10/89/8 to CNH. The funders had no role in the study design, data analysis, data interpretation, and in the writing of this report. All authors had full access to the data in the study and accept responsibility to submit for publication.

## Author contributions
M.H., G.K., C.N.H., M.P., D.S., M.S., S.W., and G.Z. designed the study and contributed to experimental design and data interpretation. D.H., J.K., H.L., S.R., Y.T., and D.W. preformed patient recruitment, clinical management, evaluation of clinical data, and sample collection. JF and LK performed bioinformatic analyses. J.A., J.B., J.F., VF, R.G., D.H., J.K., H.L., D.S., S.U., S.W., and G.Z. performed experiments and analyzed and processed the data. J.F., M.H., G.K., M.P., M.S., and G.Z. wrote the manuscript. M.H., G.K., C.N.H., M.P., M.S., and R.T. were involved in funding acquisition.

## Funding

## Competing interests
All authors declare to have no financial or other associations that might pose a potential or actual conflict of interest.
