## [Peer Review File · Nature Communications]

REVIEWER COMMENTS

Reviewer #1 (Remarks to the Author):

The authors present a clearly written and carefully conducted case study of chronic SARS-CoV-2 infection in a single immunocompromised patient, using a comprehensive range of techniques to confirm and characterise the changes occurring in the viral genome over the course of infection. A number of similar studies have been published and preprinted over the past six months, including some high profile early studies (eg Lauring, Gupta, etc), which likewise describe virus evolution within a single patient and point to the emergence of mutations of concern. The existence of previous reports limits the novelty of these findings, but the present report is important in adding to the growing body of evidence of the evolutionary potential of SARS-CoV-2, and adds substantially to the field. Additional data on differential neutralisation of the observed mutations are also useful and novel.

The case presented here shows SARS-CoV-2 infection within a patient that persisted with a genetically conserved majority viral population (identical “early” consensus genomes) until a new “late” divergent lineage arose, apparently de novo, by day 42, and became dominant by day 56. The original lineage was still detectable three days later (day 59), but the new “late” variant took over and continued to evolve until day 140, after which the patient cleared the infection following treatment changes. Although anti-nucleoprotein(N) antibodies remained detectable throughout the infection (from day 12), an appreciable level of anti-Spike (S1-subunit)-specific antibodies was only detected after day 140, coinciding with the initiation of viral clearance. Substantial differences are shown between the “early” and “late” (d105) genotypes in replicative fitness and capacity for immune escape, with both genotypes inducing protective immunity in a mouse model, and no evidence of T cell escape.

The report is detailed, thorough, and well written. The figures are also very clear. My only general comment is that it is not straightforward for the reader to glean from the abstract that what is observed is primarily the diversification of a single viral population (Fig 2a). The abstract (lines 31-33) says “several virus variants emerged”, and the text throughout refers to “various variants” – but the data presented show multiple mutations arising, leading to the emergence two (?) late viral populations (with the d105 one exhibiting reduced fitness in culture and in a mouse model). The word “variants” has unfortunately been used in recent publications to mean both individual mutations and whole-genome haplotypes, and in this case it would be useful to clarify.

Specific comments

- Abstract lines 37-39 – what new avenues for updating vaccines do you see being opened up by this work?
- Methods – there is mention of confirmatory Sanger sequencing of Spike (line 121) and references to co-occurrence of deletions and SNVs later in the text, but no Sanger sequencing in the methods. Please clarify.
- What was the phylogenetic model selected by IQ-TREE after model test?
- Why were different variant cutoffs used (Fig 2 v Fig 4?) What was the basis for choosing these thresholds?
- Fig 2A – For many readers it would be more informative to indicate the Pango lineages as well as the nextclade designations.
- Fig 2A – Please also indicate all consensus-level mutations that fall on the long branches leading to the “late” consensus genomes, ie the new lineage-defining mutations (this can be done using ancestral state reconstruction in IQTREE, although in this case is also easy to do manually)
- Fig 2C – I struggled to distinguish the threshold-passing variants in the heatmap. Is it possible to mark (perhaps with a dot or cross over the relevant heatmap cell) the variants where the frequency exceeds the given thresholds of 10 and 25%?
- Fig 4C – again, difficult to identify the threshold-passing variants, please mark these

Reviewer #2 (Remarks to the Author):

Viral evolution in immunocompromised patients is a very important question, and the authors are to be commended on their work. Major limitation remains that this is data presented on a single patient, and generalizability is unclear. Additionally, correlations with degree of immunosuppression are not clear.

SPECIFIC COMMENTS

Case presentation:

- What induction immunosuppression did patient receive for transplant?
- Were antivirals administered as part of clinical trial? E.g. why was ivermectin and remdesvir not administered until day 56 and day 140, respectively.

- By “kidney transplant failure,” do authors mean rejection or acute kidney injury. If rejection, how was it treated? Would be important to note any augmentation in immunosuppression

Figure 3: Any theorized conformational changes based on mutations?

Figure 4: Are there viral strain controls for N- and spike proteins IFE and WB?

Discussion

- Authors theorize that lack of development of spike antibody contributed to development of variants. Do the authors have thoughts on the role of antiviral therapy (e.g. decreasing viral load) and degree of immunosuppression on viral evolution?

- Do authors think that potential limitation in isolation d105 virus (e.g. genome alterations) may have had downstream effects on all further assays?

Reviewer #3 (Remarks to the Author):

In this work, Weigang et al. examine intra-host evolution of SARS-CoV-2 in an immunosuppressed kidney transplant recipient. They find that the virus gains substitutions in spike such as the E484G and deletions 141-144 and 244-247 in spike. Similar deletions occur in B.1.1.7 (del 141) and B.1.351 (del 241-243) and mutations in 484 (mostly E484K or E484Q) occur in multiple variants of concern. While intra-host evolution has been previously characterized, the authors go further and outgrow the virus to show: 1) Escape of the evolved virus from neutralization by the patient plasma and, to a lesser extent, from plasma from convalescents with moderate disease or vaccination; 2) attenuated growth dynamics and virulence of the intra-host evolved virus; 2) Mostly the deletions mediating escape from convalescent plasma, with some contribution of E484G in addition to the deletions in decreasing neutralization of Pfizer BNT162b2 elicited plasma (the last results using a pseudo-neutralization system). Furthermore, they show that infection of mice with the evolved virus elicits plasma which can neutralize B.1.351. The authors suggest that one implication of these results is that variants of concern may arise by intra-host evolution.

While this is a highly significant paper, there are major weaknesses, especially with the day 105 isolate, which has acquired changes during in vitro outgrowth (see point 1 below). To address them, the authors would need to redo a great deal of experimentation. After careful consideration and

given that there is no guarantee that some of the changes in the virus can be avoided, I would recommend that the authors qualify their results rather than repeat them.

1) The outgrown day 105 evolved virus differs from what was originally sequenced in the swab in important ways: a) The F490L mutation is lost; b) the E484G mutation is fixed; c) the furin cleavage site is deleted. These in vitro changes which are the consequence of the outgrowth may be sufficient to attenuate the virus, and therefore the attenuation which the authors observe may be an artifact of the outgrowth, not a feature of the original day 105 virus. From my understanding of the methods, the authors used VeroE6 cells to outgrow the virus, and the VeroE6 outgrowth would introduce the furin cleavage site deletion, and perhaps contribute to the other changes. While the day 14 isolate does not have these problems, the differential sensitivity of more evolved variants to the furin cleavage site mutations/deletion has been previously observed. The furin cleavage site mutants can be cleared with CaLu-3 cell passaging, but there is no guarantee that they can outgrow the virus without losing F490L and fixing E484G. Therefore, the attenuation data should be carefully and clearly qualified in addition to what is already done in the paper.

2) It is difficult to understand what selective pressure caused the day 105 virus to evolve. According to the ELISA data, antibodies to spike did not appear until day 140. The authors show that there was no neutralization of day 14 isolated virus on day 123. I do not think their observations are wrong, but this fact should be highlighted – that evolution of escape occurred before there was measurable selective pressure for it, indicating that either the selective pressure was compartmentalized to the tissues or that the virus evolved for a different reason.

3) The authors state in the abstract that: “Importantly, infection of susceptible hACE2-transgenic mice with one of the patient’s escape variants elicited protective immunity against re-infection with either the parental virus, the escape variant or the South African variant of concern, demonstrating broad immune control.” I am struggling to see the basis for this statement. From figure 6a-e, the day 105 virus-elicited plasma neutralized the day 105 virus and B.1.351, with reduced neutralization of day 14 virus and B.1.1.7. Day 14 virus-elicited plasma neutralized the day 14 virus and B.1.1.7, with reduced neutralization of day 105 virus and B.1.351. It seems from this that the breadth of the day 14 versus 105 elicited plasmas is similar, but the variants they can neutralize are different. For clarity, the authors should show the geometric mean of the fold-decrease in neutralization for each pair on the graph, and if the breadth is similar, their conclusion should be modified.

4) The results showing that the F490L mutation and to a lesser extent the E484G mutation (which do not appear on variants of concern) may not have a strong involvement in antibody escape may mean that they evolved for other reasons. Furthermore, if the evolved virus is attenuated as the authors suggest, it might not be able to spread well. Therefore, the statement in the abstract that “immunocompromised patients are an alarming source of potentially harmful SARS-CoV-2 variants” needs to be qualified.

AW: Nat. Commun. NCOMMS-21-15866-T, Decision letter of 25.05.2021

Point-by-Point-Response to reviewers:

We would like to thank the reviewers for their positive evaluation of our manuscript,
Changes are highlighted in the corrected –compare-copy- version.
The line numbers refer to the revised, clean version of the manuscript.

Reviewer #1:

1.1. My only general comment is that it is not straightforward for the reader to glean from the abstract that what is observed is primarily the diversification of a single viral population (Fig 2a). The abstract (lines 31-33) says “several virus variants emerged”, and the text throughout refers to “various variants” – but the data presented show multiple mutations arising, leading to the emergence two (?) late viral populations (with the d105 one exhibiting reduced fitness in culture and in a mouse model). The word “variants” has unfortunately been used in recent publications to mean both individual mutations and whole-genome haplotypes, and in this case it would be useful to clarify.

- We rephrased the wording in the abstract and throughout the main text. We now use the term “variant” only for the virus isolate with a well-defined genotype and the term “mutations” to describe genomic changes in the viral populations detected in the oral swabs sampled at different time points from the persistently infected patient. (lines 31-36, 63-66, 127-128)

1.2 Abstract lines 37-39 – what new avenues for updating vaccines do you see being opened up by this work?

- The analysis of the mouse sera from d105-infected animals shows a clear shift to an increased neutralization of the d105 virus as well as the beta variant of concern, suggesting that introduction of such mutations/amino acid exchanges into the spike sequence of the current COVID-19 vaccine might broaden its effectiveness against emerging SARS-CoV-2 variants. (This is now mentioned, discussion, line 302-303).

1.3 Methods – there is mention of confirmatory Sanger sequencing of Spike (line 121) and references to co-occurrence of deletions and SNVs later in the text, but no Sanger sequencing in the methods. Please clarify.

- We now describe Sanger sequencing of the spike cDNAs in the Methods part (lines 711-719).

1.4 What was the phylogenetic model selected by IQ-TREE after model test?

- The phylogenetic model was GTR+F+I and so far only mentioned in the figure legend of figure 2. This has now been added to the methods part (line 726).

1.5 Why were different variant cutoffs used (Fig 2 v Fig 4?). What was the basis for choosing these thresholds?

- We used different cutoff levels in the sequence comparisons in figure 2c and 4a in order to not exceed the size of the figures. Because some mutations in the spike gene appeared at lower frequencies in the swab samples, we used a cutoff of 10% for the spike gene in Fig. 2c, indicated at the right of the figure. In addition, we now show the full data of the variant calling with an uniform variant cutoff of 10 % for both figures in supplementary figure 1 and 2a.

1.6 Fig 2A – For many readers it would be more informative to indicate the Pango lineages as well as the nextclade designations.

- Due to the diversity of the pangolin nomenclature, in multiple cases only one viral sequence belongs to one specific pangolin lineage. Therefore, the plot would become confusing and the phylogenetic diversification would not be as easy to follow. As an example shown below, we plotted this for the Germany tree but do not like to include this information into the manuscript.

1.7 Fig 2A – Please also indicate all consensus-level mutations that fall on the long branches leading to the “late” consensus genomes, ie the new lineage-defining mutations (this can be done using ancestral state reconstruction in IQTREE, although in this case is also easy to do manually).

- As the reviewer suggested, we reconstructed the ancestral sequences with IQTREE for each node and compared these with the respective consensus sequences, excluding INDELS. These mutations are now displayed for the late consensus genomes in Figure 2a (line 728-729).

1.8 Fig 2C – I struggled to distinguish the threshold-passing variants in the heatmap. Is it possible to mark (perhaps with a dot or cross over the relevant heatmap cell) the variants where the frequency exceeds the given thresholds of 10 and 25%?

- We now provide the full dataset of Fig 2c and Fig 4a in the supplementary figure 1. Here, we plotted all variant frequency values in the respective cells.

1.9 Fig 4C panel a – again, difficult to identify the threshold-passing variants, please mark these.

- This is now addressed in supplementary figure 2a (see comment 1.8).

Reviewer #2: Case presentation:

2.1 What induction immunosuppression did patient receive for transplant?

- We now specify the induction of immunosuppression. (Methods: Case history, line 553-559) (see also reviewer 3.5)

2.2 Were antivirals administered as part of clinical trial? E.g. why was ivermectin and remdesvir not administered until day 56 and day 140, respectively.

- Antiviral therapy was not part of a clinical trial. Ivermectin treatment (33 mg/day) was initiated since at that time no other medication was available and local guidelines of the Freiburg University Medical Center allowed this therapy for 5 days. Remdesivir treatment followed the respective approval of the European Medicines Agency (EMA) that allows treatment for a maximum of 10 days. Accordingly, the patient received the maximal allowed dosage of Remdesivir (200 mg on day 1, 100 mg/day 2 to 10). This information is now included in the Methods part (lines 624-627).

2.3 By “kidney transplant failure,” do authors mean rejection or acute kidney injury. If rejection, how was it treated? Would be important to note any augmentation in immunosuppression.

- We now specified in the Result section that it was (lines kidney transplant failure (line 79-81): In May the patient suffered from an acute kidney injury (stage 1) in his transplant due to an urinary tract infection with *E. coli*, that required antibiotic treatment. Immunosuppressive medication remained unchanged.

2.4 Figure 3: Any theorized conformational changes based on mutations?

- To our knowledge, detailed studies about possible changes in the architecture of the spike protomer or trimer by the deletions in the NTP or the exchanges in the RBD are not available. (Discussion, lines 267-269)

2.5 Figure 4: Are there viral strain controls for N- and spike proteins IFE and WB?

- We now compared the intracellular replication of the d14 and 105 variants with a prototypic Muc-IMB-1 isolate, lineage B.1, that was isolated in late January, 2020 (Wolfel et al., 2020) (line 152) by IFA and WB (Fig. 4b and c) (lines 150-152).

2.6 Discussion, Authors theorize that lack of development of spike antibody contributed to development of variants. Do the authors have thoughts on the role of antiviral therapy (e.g. decreasing viral load) and degree of immunosuppression on viral evolution?

- We speculated that the constant presence of spike specific antibodies, although oscillating around the cutoff value of the ELISA (Fig. 1f and lines 97-99) and with no detectable neutralizing activity until day 123 (Fig. 5a), might have caused the emergence of the spike mutants detected at d105 and d140 (Fig. 2c-e and Discussion lines 270-274). The degree of immunosuppression most likely hampered the development of neutralizing antibodies and therefore permitted viral evolution (Discussion lines 282-289). However, the antiviral therapy with Remdesivir could not have influenced viral evolution between day 42 to 140 because it started only on d140 to d150 (Fig. 1c).

2.7 Do authors think that potential limitation in isolation d105 virus (e.g. genome alterations) may have had downstream effects on all further assays?

- The d105 isolate, now designated d105(del) showed some genomic alterations, most important, a deletion of seven amino acids in the furin cleavage site of the spike protein that was not found in the viral sequences from the d105 swap and was most likely acquired during cell culture passage (now shown in supplementary Fig. 2a and lines 146-149). We now show that this alteration has led to the attenuation of the d105(del) mutant virus. To exclude that the deletion in the furin cleavage site is responsible for the antibody escape, we now isolated a new d105 isolate, designated d105, without changes in the furin cleavage site by plaque purification and growth on Calu-3 cells (new Fig. 4). Neutralization assays with this new d105 variant showed comparable escape from antibody neutralization as shown for the initial d105(del) isolate (supplementary Fig. 3). We confirmed these findings independently using our VSV* Δ G(FLuc)-based system (Fig. 5c to e). Based on the similar escape of both isolates d105(del) and d105 from antibody neutralization, we think that the attenuation of d105(del) did not influence the results of the respective neutralization assays.

Reviewer #3:

3.1 The outgrown day 105 evolved virus differs from what was originally sequenced in the swab in important ways: a) The F490L mutation is lost; b) the E484G mutation is fixed; c) the furin cleavage site is deleted. These in vitro changes which are the consequence of the outgrowth may be sufficient to attenuate the virus, and therefore the attenuation which the authors observe may be an artifact of the outgrowth, not a feature of the original day 105 virus. From my understanding of the methods, the authors used VeroE6 cells to outgrow the virus, and the VeroE6 outgrowth would introduce the furin cleavage site deletion, and perhaps contribute to the other changes. While the day 14 isolate does not have these problems, the differential sensitivity of more evolved variants to the furin cleavage site mutations/deletion has been previously observed. The furin cleavage site mutants can be cleared with Calu-3 cell passaging, but there is no guarantee that they can outgrow the virus without losing F490L and fixing E484G. Therefore, the attenuation data should be carefully and clearly qualified in addition to what is already done in the paper.

- We highlight in the text that the d105 virus population detected in the patient specimen consisted of a complex mixture of genome mutations (Fig. 2c, lines 123-128). At least two viral spike variants, including del141-144/F490L that got prevalent in the d140 specimen (Fig. 2c) and the del244-247/E484G variant that was found in the d105 cell culture isolate (Fig. 4a). The main alterations in the spike protein are summarized in figure 2e.

- As mentioned by the reviewer the deletion of the furin cleavage site is found in about 50% of the d105 isolated virus (now referred to as d105(del)) and most likely occurred during cell culture passage on VeroE6 cells. As suggested by the reviewer (see also reviewer 2.7), we now plaque purified a d105 variant on Calu-3 cells without that deletion (now designated d105). The new d105 variant showed no attenuation in cell culture and in vivo (new Fig. 4), but showed a comparable escape from neutralization by convalescent anti-sera (compare Fig. 5b with supplementary Fig. 3). This indicates that the attenuating loss of the furin cleavage site had no influence on the later neutralization assays. Unfortunately, the sera from the immunosuppressed patient were limited and we could not repeat the neutralization assays shown in figure 5a with the new d105 isolate. Therefore, we kept the analysis of the patient's antisera with the d105(del) variant in figure 5a and b, as in the initial submission, but moved the initial characterization of d105(del) to the supplement (supplementary Fig. 2).

3.2 It is difficult to understand what selective pressure caused the day 105 virus to evolve. According to the ELISA data, antibodies to spike did not appear until day 140. The authors show that there was no neutralization of day 14 isolated virus on day 123. I do not think their observations are wrong, but this fact should be highlighted – that evolution of escape occurred before there was measurable selective pressure for it, indicating that either the selective pressure was compartmentalized to the tissues or that the virus evolved for a different reason.

- As mentioned above (2.6) we speculate that the long term presence of spike specific antibodies constantly oscillating around the cutoff value of the S1 ELISA between day 40 to 123 (Fig. 1f), although negative in the neutralization assays (Fig. 5a), might have caused the emergence of d105 (Discussion lines 270-273).

3.3 The authors state in the abstract that: “Importantly, infection of susceptible hACE2-transgenic mice with one of the patient’s escape variants elicited protective immunity against re-infection with either the parental virus, the escape variant or the South African variant of concern, demonstrating broad immune control.” I am struggling to see the basis for this statement. From figure 6a-e, the day 105 virus-elicited plasma neutralized the day 105 virus and B.1.351, with reduced neutralization of day 14 virus and B.1.1.7. Day 14 virus-elicited plasma neutralized the day 14 virus and B.1.1.7, with reduced neutralization of day 105 virus and B.1.351. It seems from this that the breadth of the day 14 versus 105 elicited plasmas is similar, but the variants they can neutralize are different. For clarity, the authors should show the geometric mean of the fold-decrease in neutralization for each pair on the graph, and if the breadth is similar, their conclusion should be modified.

- We calculated the fold change of the neutralizing titers between the d14 and d105 isolate and between B.1.1.7 and B.1.351 of sera from wt-infected and d105-infected mice as suggested by the reviewer. For both pairs, d14 versus d105 and B.1.1.7 versus B.1.351, the sera of wildtype-infected mice showed a broad variation, with a reduced neutralization capacity for the escape variants, and geometric means of about 2.73 and 2.54, respectively. The sera of animals infected with the d105 variant showed reduced variation, with an increased capacity to neutralize the escape variants when compared to the wt viruses, with a geometric mean of 1.74 and 1.40 (see the figure below). Although, these differences are small, they support our assumption that infection with the d105 variant elicited a broad immune response against all different viral variants tested (lines 34-37, 64-66 and 290-297).

3.4 The results showing that the F490L mutation and to a lesser extent the E484G mutation (which do not appear on variants of concern) may not have a strong involvement in antibody escape may mean that they evolved for other reasons. Furthermore, if the evolved virus is attenuated as the authors suggest, it might not be able to spread well. Therefore, the statement in the abstract that “immunocompromised patients are an alarming source of potentially harmful SARS-CoV-2 variants” needs to be qualified.

- As shown in figure 2e, our analysis identified a variety of mutations, resulting in amino acid deletions and exchanges in the spike protein, that are also found in previous studies of antibody escape mutations of the spike gene of variants of concern. However, as mentioned by the reviewer the degree of involvement in antibody escape may vary. For technical reasons (isolation using VeroE6 cells) the initial d105(del) isolate was indeed attenuated. However, the new d105 isolate from Calu-3 cells displayed no deletion in the furin cleavage site and was not attenuated in cell culture and in vivo (see new figure 4). This supports our statement that mutant viruses emerging from immunocompromised patients might be an alarming source of new variants of concern (Abstract lines 38-40).

3.5 What induction immunosuppression did patient receive for transplant?

- As mentioned above (2.1) we now specify the induction of immunosuppression that started 12 days before the first positive RT-PCR result (day 0). (Methods: Case history, line 553-559)

REVIEWERS' COMMENTS

Reviewer #1 (Remarks to the Author):

Thank you; all my comments have been fully addressed in the revised manuscript. In view of recent media discourse, and in line with the suggestion made by one of the other reviewers, I would encourage the authors to consider toning down the language at the end of the abstract - particularly "alarming source of new variants" - this reads 'clickbaity', and the capacity for alarm has arguably been exhausted by this stage of the pandemic.

Reviewer #3 (Remarks to the Author):

The authors have addressed my concerns.

There are minor points which should be considered:

1) Abstract lines 36-37 "demonstrating broad immune control against such variants of concern" and Introduction lines 64-65 "late virus variant isolated from the patient elicited a broad protective immune response in experimentally infected mice, suggesting that convalescent individuals might become resistant against reinfection by emerging variants of concern"

The delta variant escapes from plasma neutralization elicited by the beta variant. The d105 virus the authors isolated is something close to the beta serotype and likely would not elicit good delta neutralization.

Also, the data for increased breadth of the d105 vs the d14 isolate is not presented in the paper but only as a reply to comment 3.3 in the rebuttal letter.

So "broad" may be inaccurate and should be qualified.

2) Re-isolating the virus in Calu-3 made the results clearer. However, why the previous d105del isolate is used in some of the experiments may not be very clear to someone who doesn't know the paper's history.

I suggest replacing "For technical reasons, the d105(del) variant was used in some of the following serological assays" on lines 164-165 with an explanation. For example, "we first outgrew the d105 in VeroE6 and because of in vitro mutations re-isolated using Calu-3. Not all work was repeated with the re-isolated virus because samples were expended, but we did confirm that neutralization was not affected by the VeroE6 introduced changes."

AW: Nat. Commun. NCOMMS-21-15866A, Decision letter of 17.09.2021

Point-by-Point-Response to reviewers:

We thank both reviewers for their second evaluation of our manuscript and their helpful suggestions. All changes are highlighted yellow in the corrected –compare-copy- version of the text. The line numbers refer to the revised, clean version of the manuscript.

Reviewer #1:

1.1. All my comments have been fully addressed in the revised manuscript. In view of recent media discourse, and in line with the suggestion made by one of the other reviewers, I would encourage the authors to consider toning down the language at the end of the abstract - particularly "alarming source of new variants" - this reads 'clickbaity', and the capacity for alarm has arguably been exhausted by this stage of the pandemic..

- We rephrased the wording in the abstract (lines 38-40) according to the reviewer's suggestion. "Our results suggest indicate that immunocompromised patients could be a an alarming source of potentially harmful SARS-CoV-2 variants"

Reviewer #3:

The authors have addressed my concerns. There are minor points which should be considered:

3.1. Abstract lines 36-37 "demonstrating broad immune control against such variants of concern" and Introduction lines 64-65 "late virus variant isolated from the patient elicited a broad protective immune response in experimentally infected mice, suggesting that convalescent individuals might become resistant against reinfection by emerging variants of concern"

The delta variant escapes from plasma neutralization elicited by the beta variant. The d105 virus the authors isolated is something close to the beta serotype and likely would not elicit good delta neutralization. Also, the data for increased breadth of the d105 vs the d14 isolate is not presented in the paper but only as a reply to comment 3.3 in the rebuttal letter. So "broad" may be inaccurate and should be qualified.

- We revised the wording in the abstract (line 36) and the introduction (line 64) accordingly. " demonstrating a considerable broad immune control against such variants of concern."

3.2. Re-isolating the virus in Calu-3 made the results clearer. However, why the previous d105del isolate is used in some of the experiments may not be very clear to someone who doesn't know the paper's history.

I suggest replacing "For technical reasons, the d105(del) variant was used in some of the following serological assays" on lines 164-165 with an explanation. For example, "we first outgrew the d105 in VeroE6 and because of in vitro mutations re-isolated using Calu-3. Not all work was

repeated with the re-isolated virus because samples were expended, but we did confirm that neutralization was not affected by the VeroE6 introduced changes.”

- We thank the reviewer for the suggestion and changed the text accordingly:

(line 165) ~~„For technical reasons, the d105(del) variant was used in some of the following serological assays.~~

(now in lines 181-186): „Because of the mutation in the furin cleavage site of the d105(del) spike protein, we re-isolated d105 without changes in the furin cleavage site using Calu-3 cells. This new d105 isolate showed a comparable escape from antibody neutralization (Supplementary Fig. 3), indicating that the eight amino acids deletion in the spike of d105(del) did not affect the neutralization sensitivity of the d105 spike variant. Of note, not all work could be repeated with this re-isolated d105 virus because samples were expended.“